# The Architectural Layout of Long-Term Care Units: Relationships between Support for Residents’ Well-Being and for Caregivers’ Burnout and Resilience

**DOI:** 10.3390/ijerph21050575

**Published:** 2024-04-30

**Authors:** Yifat Rom, Ido Morag, Yuval Palgi, Michal Isaacson

**Affiliations:** 1Department of Gerontology, University of Haifa, Abba Khushi Ave. 199, Haifa 3498838, Israel; ypalgi@research.haifa.ac.il (Y.P.); michali@mit.edu (M.I.); 2Shenkar College of Engineering and Design, School of Industrial Engineering and Management, Ramat-Gan 5252626, Israel; ido-ilit@013.net.il; 3MIT AgeLab-Massachusetts Institute of Technology, Cambridge, MA 02142, USA

**Keywords:** special dementia units, caregivers’ resilience and burnout, long-term care, nurse–patient relationships, occupational health

## Abstract

With a growing need for long-term care facilities in general, and for specialized dementia units in particular, it is important to ensure that the architectural layouts of such facilities support the well-being of both the residents and the unit caregivers. This study aimed to investigate correlations between the support provided by the architectural layout of long-term care units for enhancing residents’ well-being and for decreasing unit caregivers’ burnout and increasing their resilience—as layouts may impact each party differently. The Psycho Spatial Evaluation Tool was utilized to assess the support provided by the layouts of seventeen long-term care units (ten regular nursing units and seven specialized dementia units) for the residents’ physical and social well-being (five dimensions); a questionnaire was used to measure the unit caregivers’ burnout and resilience. When analyzing layouts’ support for residents’ physical and social well-being, inconsistencies emerged regarding correlations with caregivers’ burnout and resilience across the two types of long-term care units. Supporting residents’ physical well-being was correlated with increased caregiver resilience in dementia units, and with increased burnout and decreased resilience in regular nursing units. Layouts supporting social well-being showed inconsistent correlations with caregivers’ resilience indexes in dementia units, and with burnout and resilience indexes in regular nursing units. The findings underscore the role of the architectural layout of long-term care units in enhancing residents’ well-being; the results also highlight the possible unintentional yet negative impact of the layout on the caregivers’ burnout and resilience. This study emphasizes the need to identify and rectify design shortcomings as a means of enhancing residents’ well-being, while increasing the unit caregivers’ resilience and decreasing their burnout. These insights should be addressed when developing strategies and interventions for ensuring optimal care environments for all parties involved.

## 1. Introduction

The desire to design long-term care facilities that optimize their residents’ well-being and the quality of received care has led to an abundance of studies on the correlations between multidisciplinary design variables, the residents’ well-being, and the caregivers’ burnout and resilience [1,2,3]. Specifically, the following four aspects are often addressed in the literature: (1) environmental variables, e.g., noise levels, odors, natural scenery, and levels of crowding; (2) design variables that enhance various aspects of residents’ well-being, e.g., comfort, privacy, wayfinding, security, and safety [4,5,6]; (3) residents’ ability to engage in social interactions with unit caregivers, fellow residents, and visitors [7,8]; and (4) type of long-term care units. The latter is especially important since the Ministry of Health in several countries requires the categorization of residents into regular nursing units (RNU) or specialized dementia units (SDU), for example, based on their physical dependency levels. RNUs house physically dependent residents who require assistance with daily tasks; SDUs, on the other hand, house physically independent residents who are experiencing a severe cognitive decline combined with various behavioral symptoms, and as such require around-the-clock supervision.

The well-being of the residents and the burnout and resilience of the unit caregivers (nurses and nurse aids)—who spend long shifts in close proximity to the residents—are intertwined and have a heightened reciprocal impact [9,10]. For example, disruptive behaviors among residents may stem from needs that are not met by the caregivers; in turn, this could trigger negative responses in the caregivers, increasing burnout or apathy and leading to the further worsening of the residents’ disruptive behaviors [11]. As such, just as architectural layout variables are incorporated to support the residents’ well-being, they should also offer support for the unit caregivers’ decreased burnout and increased resilience—for the benefit of all parties involved.

As with previous age-related well-being studies [12], this research adopts the Social Production Function Model [13], whereby well-being is conceptualized as a universal objective, attainable through the satisfying of needs, and may be measured through five instrumental dimensions: comfort; stimulation; status; behavioral confirmation; and affection. These goals can be supported by both physical and non-physical resources, referred to as the “means of production”. According to the Social Production Function Model, these five instrumental dimensions and their associated resources contribute to the individuals’ physical well-being (PWB) and social well-being (SWB). Moreover, these components have symbiotic relationships, with individuals turning to the available resources to compensate for deficiencies, especially diverse age-related losses [14].

The levels of burnout and resilience among unit caregivers have been found to be correlated with their well-being [15], which is assessed across four key domains: (1) job satisfaction, including attitudes towards salaries, professional development, and promotion opportunities [16]; (2) fatigue, following the residents’ deterioration and eventual demise, including psychological, emotional, and physical fatigue (e.g., muscular strain from extensive standing, walking, and lifting) [17]; (3) occupational stress, which stems from negative interactions with colleagues, residents’ families, and the residents themselves (e.g., conflicts about dressing and bathing, verbal abuse, and excessive demands) [18,19]; and (4) burnout, resulting from the caregivers’ ongoing occupational stress [20]. In general, the outcome of these four domains could have a detrimental effect on unit caregivers, including increased turnover and absenteeism, and decreased morale and motivation [21]. Specifically, burnout could impair the caregiver–resident relationship, impacting the latter’s ability and desire to provide empathic and quality care [16,22,23].

The literature contains abundant research on the architectural design aspects that harm or contribute to the well-being of residents in long-term care units or to the unit caregivers’ burnout and resilience [24,25]. However, limited attention has been given to the simultaneous effects of these design aspects on both parties [26,27]. Moreover, studies on caregivers’ fatigue often focus on functional aspects of the architectural layout, with an emphasis on spatial adjacencies, visual connectivity, room size, bedroom standardization, and restorative break areas. Design aspects relating to the residents’ well-being or to the caregivers’ burnout and resilience have been found to be interconnected, yet these may impact each party differently. For example, while private rooms may be desirable for the residents, they increase the size of the unit, thereby increasing the caregivers’ fatigue and decreasing their well-being, as they are required to walk greater distances during their shifts [28].

Based on this review of the literature, this study aims to examine the extent to which architectural layouts contribute to the well-being of residents across different unit types while concurrently assessing their influence on the burnout and resilience levels of caregivers within those units. Additionally, this research seeks to delineate and analyze the inherent contradictions between the needs of residents and caregivers within these units. The findings of this study could shed light on this complex relationship while providing valuable insights into how the units’ architectural layouts can support the well-being of residents, decrease the caregivers’ burnout, and bolster their resilience.

For this study, the following two research hypotheses were defined:

**H1.** 
*Correlations will be seen between layouts of SDUs and RNUs that support the residents’ well-being (physical and social) and the unit caregivers*
*’ burnout and resilience.*


**H2.** 
*Despite similarities in the layouts of SDUs and RNUs, differences will be seen in their impact on unit caregivers*
*’ levels of burnout and resilience.*


## 2. Materials and Method

In 2022, the researchers approached 20 privately owned long-term care facilities in Israel. Consent to participate in this study was obtained from five of these facilities, resulting in ten RNUs and seven SDUs (seventeen units in total). Seven inclusion criteria were applied: (1) each facility had at least one SDU and one RNU; (2) SDUs and RNUs in the same facility had identical architectural layouts; (3) each facility was designed and built in line with the Ministry of Health’s guidelines for long-term care facilities; (4) each facility was licensed by the Ministry of Health; (5) all units had a capacity of at least 25 residents; (6) researchers had access to up-to-date computerized floor plans of the units, preferably in ACAD format; and (7) the units had been continuously operating for at least six months prior to the study. In addition, a total of 340 unit caregivers (20 from each of the 17 participating units) were asked to complete a printed survey.

### 2.1. Operational Measurements

Data for this research were collected via the Psycho Spatial Evaluation Tool (PSET) [3] and the Burnout and Resilience Survey [29]. The former examined 28 variables regarding the architectural layout: 17 were extracted directly from the computerized floor plans, such as distance, area size, and number of parallel bedroom doors; 11 more abstract variables were achieved through algorithms based on space syntax, such as choice, intelligibility, and visibility (Appendix A). Since some variables were scaled differently, normalization was conducted prior to analysis; moreover, for the sake of uniformity, variables that were negatively associated with well-being (e.g., bedroom visibility from the main entrance) were multiplied by (−1). PSET scores for the five instrumental dimensions of well-being needs were calculated by summing and averaging a subset of the 28 normalized layouts.

The Burnout and Resilience Survey has been employed by the Ministry of Health in Israel since 2017 [29]. The survey comprises two sections. The first part consists of fifteen questions (eight background questions and seven occupational questions). The second part consists of forty-one questions and statements from seven discrete sections. Participants were asked to rate the following items on a Likert-like scale from 1 (never) to 7 (always): (1) burnout index, in line with the Shirom–Melamed Burnout Measure Questionnaire [30] (10 items); (2) work environment index (11 items); (3) occupation index, regarding job satisfaction (8 items); (4) workplace support index, regarding relationships with employees and co-workers, and organizational commitments (7 items); and (5) well-being index, regarding the health-related well-being of the unit caregivers (5 items). The researchers uploaded the data from each survey onto a computerized system, while assigning unique codes for each respondent, unit type, and facility.

### 2.2. Design and Procedure

The current study employed a cross-sectional ecological study design, with data gathered through three sequential steps: (1) Converting the architectural features into measures regarding support for the residents’ well-being. An analysis of the computerized floor plans for each SDU and RNU using the PSET resulted in two sets of quantitative measures: the degree to which the architectural layout is likely to support the residents’ well-being (physical and social), and the extent to which it is likely to support the residents’ five well-being needs. (2) Measuring levels of burnout and resilience, computed through seven indexes: five for burnout and resilience (burnout, work environment, occupation, workplace support, and well-being), one for the overall score index of these five indexes, and one for the caregiver-resident relationship. The latter was based on the average scores of two survey items relating to the caregivers’ subjective assessments of their compromised caregiver–resident relationships: “I have no strength to invest emotionally in patients or other employees” and “I feel that I cannot be sympathetic to other employees or patients”. (3) Statistical analysis was conducted using SPSS v.27 [31], to examine correlations between the Burnout and Resilience Survey results and the PSET results that measure the support provided by the architectural layout for the residents’ well-being. The study was approved by the Research Ethics Committee at the authors’ affiliated academic institution Haifa University, Israel and conducted in line with ethical and regulatory guidelines (approval # 140/22). Confidentiality was assured to all participants.

### 2.3. Descriptive Analysis and Statistics

The PSET yielded two outcomes: (1) five numerical measures for each plan, depicting the expected level of support for the residents’ five dimensions of needs (Figure 1); and (2) two numerical measures for each plan, depicting the expected level of support for the residents’ PWB (based on comfort and stimulation) and SWB (based on status, behavioral confirmation, and affection).

Of the potential 340 unit caregivers in all participating units, 126 (37%) completed the survey (88 females aged 21–63 years), including 62 from RNUs and 64 from SDUs. More than 84% of the participants were born in Israel and working at the facility was their primary job. Table 1 presents the demographic data of the two groups of participants.

## 3. Results

Three unadjusted associations with the unit caregivers’ burnout and resilience indexes were explored: (1) unit caregiver demographics and background variables; (2) PSET measurements of the layouts’ support for residents’ five well-being needs and for their PWB and SWB; and (3) architectural layout variables.

### 3.1. Demographics and Background Characteristics and Survey Indexes

The results revealed significant correlations between the demographic and background variables and the caregivers’ burnout and resilience indexes (Table 2).

*t*-tests were conducted to assess the impact of demographic and background variables on unit caregivers’ burnout and resilience indexes, categorized by unit type. The results revealed several associations (Figure 2), indicating that the correlations between unit caregivers’ background variables and their burnout and resilience indexes differ significantly between the two types of units, with a more significant impact on the unit caregivers’ burnout and resilience being seen in SDUs than in RNUs.

#### 3.1.1. Correlations between Background Characteristics and the Seven Survey Indexes in SDUs

When analyzing the survey data regarding SDUs, all survey indexes—except for the well-being index—were significantly correlated (positively or negatively) with their unit caregivers’ background variables. In addition, working in multiple jobs was found to be significantly associated with most indexes.

Burnout index (Figure 2A). Unit caregivers with higher tenure in the profession (≥6 years) reported higher levels of burnout than their counterparts with less tenure [t_(63)_ = −2.148, *p* < 0.05].

Work environment index (Figure 2B). Those who solely worked in the unit exhibited a higher work environment index (i.e., lower resilience) than those who had additional places of employment [t_(58.81)_ = −2.99, *p* < 0.01].

Occupation index (Figure 2C–E). A higher index (i.e., lower resilience) was found to be influenced by three background variables: (a) single workplace [t_(62)_ = −3.729, *p* < 0.001]; (b) working night shifts [t_(62)_ = −2.712, *p* < 0.01]; and (c) working part-time [t_(62)_ = 2.532, *p* < 0.05].

Workplace support index (Figure 2F–H). Similar background variables as the occupation index were found to impact the workplace support index: (a) single workplace [t_(62.96)_ = −5.054, *p* < 0.001]; (b) working night shifts [t_(63)_ = −2.655, *p* < 0.05]; and (c) working part-time [t_(63)_ = 2.187, *p* < 0.05].

Caregiver-resident relationship index (Figure 2I). A higher index (i.e., compromised relationship) was seen in caregivers who only work in the unit [t_(63)_ = −3.169, *p* < 0.01].

Total index (Figure 2J–L). A higher index (i.e., lower resilience) was seen for the same three background variables as in the occupation and workplace support indexes: (a) single workplace [t_(63)_ = −4.097, *p* < 0.001]; (b) working night shifts [t_(63)_ = −2.655, *p* < 0.05]; and (c) working part-time [t_(63)_ = 2.448, *p* < 0.05].

#### 3.1.2. Correlations between Background Characteristics and Survey Indexes in RNUs

When analyzing the survey data from the RNUs, only the unit caregivers’ age and being a registered nurse were found to significantly impact the survey indexes.

Occupation index (Figure 2M). Nurse aids reported a higher occupation index (i.e., lower resilience) than registered nurses [t_(59)_ = 3.177, *p* < 0.01].

Well-being index (Figure 2N). A lower well-being index (i.e., higher resilience) was seen in older participants (r = −0.309, *p* > 0.05).

### 3.2. Architectural Layout Support for Residents’ Well-Being and the Survey Indexes

Two correlations were examined between the units’ layout support for the residents’ well-being and the surveys’ indexes: (a) support for residents’ PWB and SWB and (b) support for the residents’ five well-being needs (Table 3). Pearson’s correlation matrices were calculated separately for each unit type.

#### 3.2.1. Correlations between Layouts’ Support for PWB and SWB and Survey Indexes

Unit type was found to play a significant role in correlations between the layouts’ support for PWB and SWB and the survey indexes. With SDUs, no correlations were seen between the layouts’ support for SWB and the survey indexes; significant negative correlations were seen between the layouts’ support for PWB and the following indexes: occupation, workplace support, overall score, and caregiver–resident relationships. No significant correlations were seen for the burnout, work environment, or well-being indexes. Conversely, in the RNUs, positive correlations were only seen between the layouts’ support for PWB and the unit caregivers’ well-being, whereas strong negative correlations were seen between the units’ support for SWB and all survey indexes.

To further understand the correlations between the layouts’ support for each of the five well-being needs and the survey indexes, an analysis was conducted separately for each unit type.

#### 3.2.2. Correlations between Layouts’ Support for Comfort and Stimulation and Survey Indexes in SDUs

Negative correlations were seen between the layouts’ support for comfort and stimulation (PWB needs) and the surveys’ indexes, indicating support for both residents and unit caregivers. Regarding comfort, significant negative correlations were seen between the layouts’ support for comfort and the following four indexes: occupation, workplace support, overall survey score, and caregiver–resident relationships, whereby the greater the support, the greater the resilience, and the better the caregiver–resident relationship. However, no correlations were seen between burnout, work environment, and the well-being indexes. Regarding stimulation, significant negative correlations were seen between the layouts’ support for stimulation and the same four indexes. While an additional correlation was seen with the well-being index, no correlations were seen between the layouts’ support for stimulation and the burnout or work environment indexes.

#### 3.2.3. Correlations between Layouts’ Support for Status, Behavioral Confirmation, and Affection and Survey Indexes in SDUs

Unlike the layouts’ support for comfort and stimulation in SDUs, inconsistent correlations were seen for the level of support for status, behavioral confirmation, and affection (SWB needs). Regarding status, the layouts’ support for the residents’ status was significantly correlated with the same four indexes as comfort (i.e., occupation, workplace support, overall survey scores, and caregiver–resident relationship indexes), but in a positive manner. This indicates a contradiction between the layouts’ support for the residents’ well-being and that of the unit caregivers, which may lead to poorer caregiver–resident relationships. However, no correlations were seen with the burnout, work environment, or well-being indexes.

Regarding behavioral confirmation, the layouts’ support for the residents’ behavioral confirmation in SDUs was negatively correlated with the occupation and caregiver–resident relationship indexes, indicating that these layouts support both residents and unit caregivers. No correlations were seen with other indexes. Regarding affection, the layouts’ support for the residents’ affection in SDUs was negatively correlated with the work environment, well-being, and overall score indexes, indicating that the layouts support both residents and caregivers. No correlations were seen with other indexes.

Finally, the layouts’ support for comfort, stimulation, behavioral confirmation, and affection were found to be positively correlated with support for the residents’ well-being and for the caregivers’ levels of burnout and resilience; however, negative correlations were seen for the layouts’ support for status.

#### 3.2.4. Correlations between Layouts’ Support for Comfort and Stimulation and Survey Indexes in RNUs

For the RNUs, all correlations between the layouts’ support for comfort and stimulation and the surveys’ indexes were positive, indicating contradictions between support for residents and support for caregivers. Regarding comfort, contrary to the negative correlations in SDUs, positive correlations were observed between the layouts’ provision of comfort, burnout, and the well-being index, whereby the higher the support for comfort, the higher the burnout index and the lower the well-being index. No correlations were seen with any other indexes. Regarding stimulation, positive correlations were only seen between the layouts’ support for stimulation and the well-being index. Again, no correlations were seen with any other indexes.

#### 3.2.5. Correlations between Layouts’ Support for Status, Behavioral Confirmation, and Affection and Survey Indexes in RNUs

All correlations between the layouts’ support for status, behavioral confirmation, and affection and the surveys’ indexes presented negative correlations, indicating support for both residents and unit caregivers. Regarding status, the layouts’ support for the residents’ status exhibited negative correlations with the burnout and well-being indexes, indicating that higher support for residents’ status was correlated with decreased unit caregivers’ well-being index and increased burnout. Notably, the same two indexes were found to be positively correlated with the layouts’ support for comfort, indicating a contradiction between the RNU layouts’ support for both needs and their impact on unit caregivers’ burnout and resilience. No correlations were seen with any other indexes. Regarding behavioral confirmation, negative correlations were seen between the layouts’ support for residents’ behavioral confirmation and the following five indexes: burnout, work environment, workplace support, overall survey score, and caregiver–resident relationships. No correlations were seen with the occupation or well-being indexes. Thereby, higher support for behavioral confirmation was found to be related to lower burnout, higher resilience, and better caregiver–resident relationships. Regarding affection, negative correlations were seen with all survey indexes, indicating that support for affection is related to lower burnout, higher resilience, and better caregiver–resident relationships.

### 3.3. Architectural Layout Variables and Survey Indexes

A correlation analysis was conducted to assess relationships between the 28 architectural design variables and indicators of unit caregivers’ burnout and resilience (results are available upon request). The results revealed numerous and varied correlations, demonstrating differences in the quantity and type of relationships. Once again, in RNUs, more correlations were seen between burnout and resilience levels and the 28 architectural layout variables compared to SDUs. For instance, in RNUs, 18 correlations were seen with the burnout index, and 100 correlations were seen with the resilience index variables. In comparison, in SDUs, only eight correlations were seen with the burnout index and 73 correlations were seen with the resilience index variables. These differences underscore the need for further investigation as a means of yielding more targeted results, informing design guidelines and specific interventions, and enhancing outcomes for all parties involved. Moreover, these results are not conducive to the current research, given the study’s objective to explore potential discrepancies between the support provided by the architectural layout of the long-term care facility for the residents’ well-being and their impact on the unit caregivers’ burnout and resilience. However, they may be significant in predicting burnout and resilience and could be further examined and addressed in future research.

### 3.4. Main Contributors to the Unit Caregiver Survey Indexes in SDUs and RNUs

A regression analysis of a comprehensive dataset was conducted to discern the most influential factors that impact the unit caregivers’ burnout and resilience, categorized by the type of unit according to the three unadjusted associations: (1) demographic and background variables of unit caregivers obtained from surveys (15 variables); (2) the degree of support provided by layouts for the five well-being needs (5 variables); and (3) architectural layout variables, utilized in conjunction with the PSET framework to assess support for the five well-being needs (28 variables). Each index was examined individually, with a regression analysis conducted separately for each unit. Additional details and Appendix A appear in Appendix A and/or are available upon request.

#### 3.4.1. Main Contributors to the Unit Caregiver Survey Indexes in SDUs

In SDUs, burnout and resilience among unit caregivers were only partially predicted by their background variables and by the PSET measurements of the layouts’ support for the residents’ well-being. Nevertheless, individual architectural layout variables—such as the degree of choice in formal public areas/rooms, adjacent spaces (space syntax), and the calculated distance from bedroom doors to formal public areas/rooms—emerged as primary predictors of caregiver burnout and resilience. Only two background variables of caregivers (subjective financial situation and employment status), along with the support provided by the layouts for three specific well-being needs of residents (comfort, stimulation, and behavioral confirmation), exclusively predicted the work environment and workplace support indexes.

#### 3.4.2. Main Contributors to the Unit Caregiver Survey Indexes in RNUs

Conversely, within RNUs, the degree of support offered by architectural layouts for residents’ well-being emerged as a significant predictor of burnout and resilience among caregivers. Notably, the layouts’ support for behavioral confirmation and status, which also reflects the level of support for SWB, consistently emerged as a predictor across various indexes. Furthermore, each of the 28 architectural layout variables predicted at least one of the burnout and resilience indexes, totaling 84 predictions. Remarkably, background variables exhibited a diminished role in predicting caregiver well-being outcomes. Factors such as full-time employment and age were found to predict the burnout index. In contrast, the resilience index, particularly workplace support and the work environment, was found to be influenced by full-time employment in a single workplace and by subjective financial circumstances.

## 4. Discussion and Implications

This study is unique as it utilizes the PSET, which is rooted in the well-established Social Production Function Model [13]; the research is also designed to quantitatively measure the extent to which the architectural layouts of long-term care facilities provide support for the well-being of both their residents and their unit caregivers (with the latter being measured through levels of burnout and resilience). Consistent with the study’s hypothesis (H1, H2), the findings demonstrate correlations between the support provided by the layouts of both SDUs and RNUs for residents’ PWB and SWB and the unit caregivers’ burnout and resilience. However, these correlations do not consistently align across all variables. Notably, the findings demonstrate no correlation between SDU layouts’ support for SWB needs and the caregivers’ burnout and resilience indexes, and only a weak correlation between the RNU layouts’ support for PWB and the caregivers’ burnout and resilience indexes. Nevertheless, the correlation with the five well-being needs, comfort stimulation status, behavioral confirmation, and affection generates more detailed results.

### 4.1. Layouts’ Support for Comfort and Stimulation and the Unit Caregivers’ Burnout and Resilience Index

Consistent with the second hypothesis (H2), correlations between the layouts’ support for comfort and stimulation (which constituted the residents’ PWB) and the caregivers’ burnout and resilience indexes differed by unit type, possibly stemming from differences in the caregivers’ scope of work, due to factors such as the residents’ level of physical dependency and cognitive abilities [22]. In SDUs, comfort and stimulation were negatively correlated with most burnout and resilience indexes. Conversely, in RNUs, comfort and stimulation were only positively correlated with two burnout and resilience indexes (the burnout index and the well-being index). These findings may be explained by the positioning of the bedrooms (e.g., in relation to malodorous or visual disruptions), staff bases, and dayrooms, which could streamline the caregivers’ workflow and supervision, while increasing residents’ orientation and overall positive stimulation [32]. More optimal arrangements should incorporate shorter walking distances, enhanced visibility from the bedroom door to the staff base and vice versa, as well as partial visibility from the bedroom door to the dayroom.

### 4.2. Layouts’ Support for Status, Behavioral Confirmation, Affection, and the Unit Caregivers’ Burnout and Resilience Index

The well-being needs that constitute SWB were also examined. Contrary to the second hypothesis (H2), both in SDUs and RNUs, the layouts’ support for behavioral confirmation and affection exhibited the most prominent negative correlations with the residents’ well-being and with the unit caregivers’ burnout and resilience, irrespective of the caregivers’ scope of work; this included residents’ level of physical dependency and cognitive abilities. On the other hand, consistent with the second hypothesis, the correlation between the layouts’ support for residents’ status and the burnout and resilience indexes was inconsistent.

Status, a multifaceted and subjective concept, gauges one’s relative position within a hierarchy, primarily determined by control over limited resources [33]. Within the context of the current research, the status of SDU residents is a complex phenomenon, impacted by spatial hierarchies and architectural designs, for example, the relative distances and visibility between the bedrooms and other crucial areas, such as the staff base, dayroom, and main entrance to the unit. The hierarchical configuration of the bedrooms and their position along a single/double corridor may also contribute to the status variable. The layouts’ support for status differs by the unit type. It demonstrates negative correlations with the burnout and resilience indexes in RNUs, suggesting decreased resilience and increased burnout, and positive correlations with those in SDUs. This discrepancy could stem from differences in the residents’ physical and mental dependencies in SDUs and in RNUs, subsequently impacting the scope of work required of the unit caregivers. In SDUs, the caregivers’ duties primarily revolve around cognitive tasks and challenges, which may manifest as physical and verbal agitation among residents [34,35]. However, in RNUs, the unit caregivers’ responsibilities entail continuous physical exertion, adherence to schedules, and teamwork [36], with such strenuous work having the potential to increase fatigue and health-related issues. Moreover, certain architectural layout variables were identified as simultaneously supporting status while potentially hindering comfort and stimulation. Therefore, higher support for status may not always enhance SDU residents’ well-being (e.g., the added distance from bedrooms to dedicated public activity rooms, the longer visual distance in the unit, floor areas that are visible from bedroom doors, and the percentage of bedroom floor areas that are visible from the main entrance of the unit).

The intricate interplay observed in the relationship between the layouts’ support for status and support for the caregivers’ burnout and resilience suggests that elevating status within these layouts may not consistently represent a viable strategy for enhancing the well-being of residents and decreasing burnout and resilience simultaneously. Instead, status could be achieved through non-architectural factors, such as interpersonal relationships, particularly with caregivers, which involve respect, recognition, and voluntary deference [2]. Additional non-architectural variables may include increased rights, privileged access to scarce resources, and exemption from certain obligations [37]. Consequently, there is a compelling need for further in-depth investigation in this domain.

Behavioral confirmation is mainly supported by better visibility throughout the unit. Such visibility enhances the caregivers’ line of sight to the residents (i.e., increasing supervision and control), as well as verbal and non-verbal confirmation (e.g., by maintaining eye contact) of residents’ adherence to societal norms and expected behaviors (e.g., minor achievements or assisting other residents) [37,38,39]. As such, the architectural layout of all types of units should strike a delicate balance between ensuring caregiver–resident visibility throughout the unit and preventing overstimulation among the residents [39,40]. This could, for example, be achieved through the strategic placing of staff bases within the unit [41].

Affection is mainly supported by generating time for person-centered social interactions, which may also decrease fatigue and promote health. This could be addressed, for example, by creating layouts that foster residents’ SWB, as suggested by the Social Production Function Model, as shown in Ref. [13]. Such layouts should enable caregivers to deliver person-centered care while cultivating warm and empathetic social relationships. They should also encourage visitation, to promote affection from family and friends. The residents’ affection initiated by guests has the potential to exacerbate the physical and mental strain on caregivers, consequently reducing their resilience and increasing the risk of burnout [42].

Fostering affection within care units involves optimizing the unit caregivers’ work efficiency while minimizing unnecessary movements and disturbances within the unit (i.e., decreasing undesirable stimulation among residents). This entails reducing distances between essential areas, such as the staff base, public rooms, support facilities, and dedicated communal spaces, while optimally positioning the staff base. On the other hand, it is discernible that overly short distances might lead to discomfort among residents due to the proximity to noise from the dayroom or staff base. The ultimate measure of success relies on the improved collective well-being of both residents and caregivers, necessitating the implementation of balanced approaches. This could be achieved, for example, by designing a semi-open kitchen next to a perpendicular cluster of service rooms, all located at the center of the unit, and preferably next to the public rooms. Additional elements that contribute to establishing an inviting environment, one that enriches visitor experiences and engagement, encompass discrete entry points. These should strategically prevent direct visibility from the unit entrance to the bedrooms, while ensuring that other areas remain in sight of the staff base. Constructive social interactions are nurtured by providing diverse public spaces for activities, such as family rooms, appropriately sized corridor ends (with a minimum dimension of 2-by-2 meters, without obstructing fire exits), and adjacent external areas such as balconies, gardens, and spacious lobbies [43].

### 4.3. Residents’ Well-Being and the Unit Caregivers’ Burnout Index

Correlations were observed between the layout’s support for residents’ well-being needs and the unit caregivers’ burnout index. In RNUs, the three needs that constitute SWB were negatively associated with the unit caregivers’ burnout, positively indicating enhanced well-being for residents and reduced burnout for unit caregivers. In contrast, when examining the three PWB needs, a positive correlation was only seen for comfort. This could be explained by the unsupportive layouts that impose more physically demanding tasks on the unit caregivers. For example, when units include private bedrooms for better privacy (comfort), this may increase the general size of the unit, in turn increasing the walking distances that must be covered by the unit caregivers during their shifts—thereby increasing fatigue and even burnout; alternatively, when designs aim to provide caregivers with better control over the unit and the residents, through high visuality, for example, this could undesirably lead to overstimulation among the residents.

Notably, in SDUs, no correlations were seen between the layout’s support for all residents’ well-being needs and the caregivers’ burnout index. These findings are in line with previous research, whereby unit caregivers in SDUs experience only low-to-moderate levels of burnout [44]. In addition, the findings indicate that burnout experienced by SDU unit caregivers may be associated with their background variables, which are unrelated to the units’ layout, such as their monthly hours of work, financial stress, and having more than one place of employment. These findings could also be attributed to the physical independence of the residents, which reduces the physical workload for unit caregivers and, consequently, increases the cognitive demands inherent in the caregiving role [38].

### 4.4. Residents’ Well-Being and the Caregiver–Resident Relationship Index

In SDUs, negative correlations that indicate better caregiver–resident relationships emerged between this index and the layouts’ support for four of the five well-being dimensions—comfort, stimulation, status, and behavioral confirmation—but not for affection. This highlights that while layouts enable the establishment of empathetic social interactions between residents and caregivers, background characteristics may have a more dominant influence. In RNUs, on the other hand, such desirable correlations, which indicate better caregiver–resident relationships, only emerged with the layouts’ support for two of the residents’ SWB dimensions, behavioral confirmation and affection, not with status and not with either of the PWB needs. These results suggest that layouts which are optimized for streamlining workflows and mitigating physical fatigue could enhance caregiver relationships. Furthermore, encouraging and increasing guest visitation may alleviate some of the supervision-related workload that is placed on unit caregivers (although this may also increase their mental strain) [40]. Such desirable improvements could be achieved by strategically locating the main entrance far away from the private bedrooms, reducing visibility into the bedrooms, distancing malodorous service areas from the main entrance and congregation spaces, and designing diverse spaces for more intimate social interactions.

### 4.5. Residents’ Well-Being Unrelated to the Resilience Indexes

All unit caregivers’ resilience indexes were found to be correlated with at least one of the three SWB needs; however, in relation to the PWB needs, correlations differed between unit type. In RNUs, only the caregivers’ well-being index was found to be correlated with comfort and stimulation, indicating the prominent role of the architectural layout in supporting the unit caregivers’ health and fatigue by reducing their physical burden. Shorter walking distances for both residents and unit caregivers, for example, could be achieved through the optimal positioning of rooms, such as the residents’ bedrooms, the nurse base, and all service rooms, while ensuring caregiver–resident visibility throughout the unit. On the other hand, in SDUs, only the work environment index was not correlated with either comfort or stimulation, suggesting that less fatigue and physical burden is caused by the layout.

Finally, additional causes of these discrepancies may be beyond the scope of the current research, highlighting the need for further investigation. However, by focusing on these discrepancies, the current study sheds light on the architectural layout that may strengthen or attenuate the residents’ well-being and the unit caregivers’ burnout and resilience—depending on the type of unit. Notably, the regression results reaffirm the importance of individual architectural layout variables in predicting the burnout and resilience of unit caregivers within SDUs. These findings, therefore, highlight the necessity for further research that delves into the nuanced effects of these variables on caregiver burnout and resilience, ultimately informing evidence-based design guidelines.

In conclusion, the architectural layouts of long-term care units should be evaluated during the design phase, with an emphasis on the types and levels of dependency of the future residents. Integrating the considerations that are presented throughout this research article offers a prospective opportunity for enhancing the overall well-being and contentment of both residents and unit caregivers.

### 4.6. Research Limitations and Future Research

The COVID-19 pandemic had a significant impact on this study, both limiting and enhancing its value. On the one hand, the pandemic decreased the number of facilities that agreed to participate in the research. However, as most participating facilities had more than one RNU and one SDU, the total number of units examined (17) may have been adequate. Moreover, COVID-related restrictions disrupted the established routines of both residents and unit caregivers, hindering routine visitations that could have positively affected the well-being of both parties (i.e., decreasing the residents’ loneliness and isolation and decreasing the unit caregivers’ workload and emotional stress). However, as the study was conducted more than 12 months after pandemic-related changes were introduced, both the residents and the unit caregivers were likely to have been familiar with the new routines, potentially mitigating the negative impact of pandemic-related disruptions.

It is worth considering this study more as a proof-of-concept, given the limited number of long-term care facilities that were included in the study. Its main objective was to demonstrate the feasibility of examining the interdependencies between the architectural layout of such facilities and the quality of care provided, examined in this case through the residents’ PWB and SWB, and through the unit caregivers’ burnout and resilience.

Further research is needed for exploring workable planning and design solutions that could provide greater support for unit caregivers (e.g., shorter walking distances and higher visibility) and for residents (e.g., dividing the unit into several wings that include a staff base, small adjacent storage, and service rooms). Additional research should also address the proximity, design, and availability of outdoor spaces, as well as additional solutions within existing units through renovations (e.g., redesigning the unit’s architectural layout, although this is an expensive and time-consuming solution), interior design (e.g., repositioning room arrangements, which will mainly affect the residents’ PWB), or introducing adjustments at the management level (by focusing on tasks and human resources) [45].

## 5. Conclusions

This study offers important insights into correlations between the architectural layouts of long-term care units, the residents’ well-being, and the caregivers’ burnout and resilience, using the quantitative PSET [3] and the Burnout and Resilience Survey [29]. As hypothesized, the layouts’ support for residents’ well-being was found to be correlated with the unit caregivers’ burnout and resilience. However, unexpectedly, not all correlations were in the same direction, as specific combinations of architectural features aiming to increase the residents’ well-being may decrease that of the unit caregivers. These disparities could be attributed to the type of care unit, as SDUs and RNUs differ in their residents’ degrees of physical dependency, thereby placing different physical workloads on the unit caregivers.

These intricacies present significant challenges for architects. Notably, the layouts’ support for behavioral confirmation among residents was the only variable that was found to be a dominant predictor of the caregivers’ increased workplace support index in both types of units, and of the burnout and resilience indexes in RNUs. Offering tailored guidelines could provide architects with a greater understanding of the architectural layout variables that impact such disparities, enabling professionals to make more informed decisions regarding potential quantifiable adjustments during the planning phase for both renovations and new construction. This approach could allow stakeholders and decision makers to identify areas where architectural designs are inadequate, while implementing compensatory non-architectural measures (e.g., assignment redistribution or staffing-level adjustments).

The importance of certain architectural layout variables in predicting the caregivers’ burnout and resilience should not be underestimated, nor should the differences in the amount and type of correlated variables. This highlights the need for further investigations that could yield more targeted results, informing design guidelines and specific interventions that strive to improve outcomes for both parties.

Finally, this research emphasizes the effectiveness of the PSET as a practical instrument for conducting comprehensive analyses of the architectural layouts of long-term care facilities. Through a systematic evaluation of floor plans, this study reveals the promise of incorporating a descriptive well-being component into quantitative inquiries of such facilities to enhance the well-being of the residents while decreasing the caregivers’ burnout and increasing their resilience.

## Figures and Tables

**Figure 1 ijerph-21-00575-f001:**
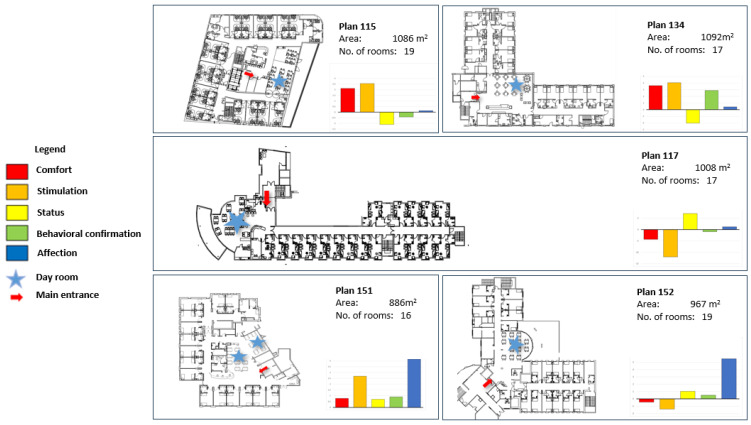
Layouts of the five long-term care facilities and levels of support for the five needs that constitute the residents’ PWB and SWB.

**Figure 2 ijerph-21-00575-f002:**
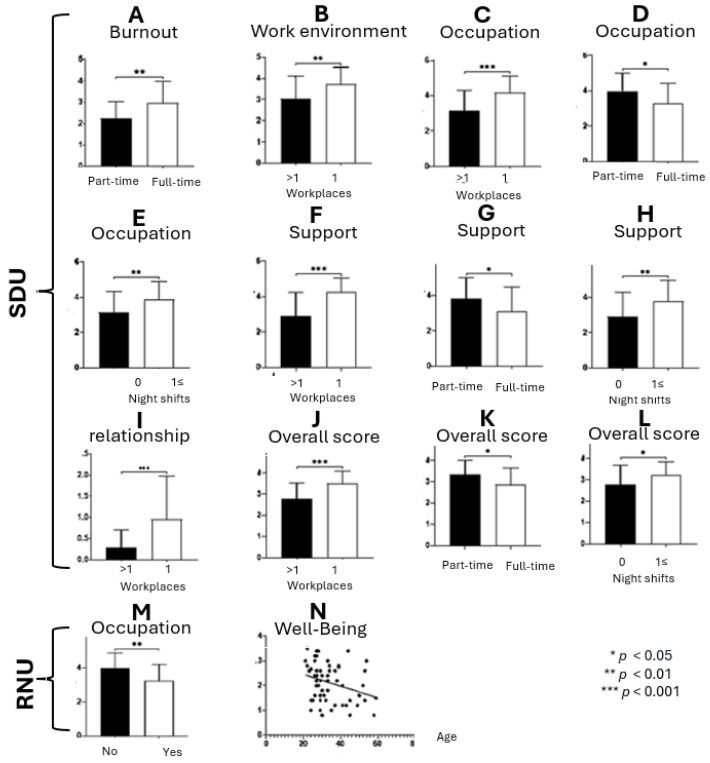
Background demographics and their impact on the unit caregivers’ burnout and resilience.

**Table 1 ijerph-21-00575-t001:** Unit caregivers’ background characteristics (*n* = 126).

Demographic Characteristics		n	%
Unit type	RNU	62	49.2
	SDU	64	50.8
Gender	Male	34	27.9
	Female	88	72.1
Registered nurse ^a^	Yes	58	44.7
	No	68	55.3
Employed in a managerial position	Yes	44	28.7
	No	82	71.3
Number of current workplaces	One	39	69.3
	More than one	88	30.7
Tenure in profession	Under six years	76	60.3
	Six years and over	50	39.7
Tenure in current facility	Under six years	95	75.4
	Six years and over	31	24.6
Monthly working hours in all workplaces	Full-time or more	92	73.0
	Part-time	34	27.0
Night shifts	Yes	74	58.7
	No	52	41.3

^a^ Statement with missing data.

**Table 2 ijerph-21-00575-t002:** Summary of the impact of background variables on the unit caregivers’ seven burnout and resilience indexes.

	Burnout	Work Environment	Occupation	Workplace Support	Well-Being	Overall Score	Unit Caregiver/Resident Relationship
Gender	-	-	-	-	+	-	-
Number of workplaces	-	−2.99 **	−3.729 ***	−5.054 ***	−2.344	−4.097 ***	−3.169 **
Tenure in the profession	−2.148 *	-	-	-	-	-	-
Tenure in the current facility	+	-	-	-	-	-	-
Full-time job	-	-	2.532 *	2.187 *	-	2.448 *	-
Night shifts	-	-	−2.712 **	−2.655 *	-	−2.267 *	-
Registered nurse	-	-	3.177 **	-	-	-	-
Managerial position	-	-	+	-	-	-	-

(+) Significance was found but cannot be interpreted due to differences in group sizes. (-) No significance was found. * *p* < 0.05. ** *p* < 0.01. *** *p* < 0.001.

**Table 3 ijerph-21-00575-t003:** Pearson’s correlations regarding the caregivers’ burnout and resilience indexes and the layouts’ support for the residents’ PWB and SWB and five well-being needs.

Unit	Parameter	Physical Well-Being			Social Well-Being			
		Total	Comfort	Stimulation	Total	Status	Behavioral Confirmation	Affection
SDU	Burnout	−0.04	−0.032	−0.044	0.075	0.054	0.051	−0.024
	Work environment	−0.207	−0.145	−0.233	−0.096	0.203	−0.091	−0.338 **
	Occupation	−0.466 **	−0.427 **	−0.480 **	0.186	0.473 **	−0.257 *	−0.207
	Workplace support	−0.378 **	−0.342 **	−0.391 **	0.103	0.394 **	−0.227	−0.232
	Well-being	−0.236	−0.171	−0.263 *	−0.156	0.234	−0.146	−0.416 **
	Overall score	−0.373 **	−0.320 **	−0.393 **	0.058	0.382 **	−0.193	−0.305 *
	Caregiver–resident relationship	−0.294 *	−0.279 *	−0.298 *	0.002	0.304 *	−0.248 *	−0.217
RNU	Burnout	0.23	0.283 *	0.204	−0.486 **	−0.283 *	−0.296 *	−0.347 **
	Work environment	−0.158	−0.048	−0.205	−0.391 **	0.079	−0.387 **	−0.642 **
	Occupation	0.075	0.123	0.053	−0.297 *	−0.069	−0.101	−0.390 **
	Workplace support	0.115	0.194	0.08	−0.452 **	−0.189	−0.324 *	−0.404 **
	Well-being	0.320 *	0.381 **	0.291 *	−0.521 **	−0.372 **	−0.206	−0.324 *
	Overall score	0.14	0.231	0.099	−0.554 **	−0.207	−0.349 **	−0.552 **
	Caregiver–resident relationship	0.071	0.134	0.042	−0.446 **	−0.135	−0.374 **	−0.448 **

* *p* < 0.05. ** *p* < 0.01.

## Data Availability

The research data, analytic methods, or materials are available to other researchers for replication purposes at yifatrom@gmail.com. The studies reported in the manuscript were not pre-registered.

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
