# Peer review of "The Architectural Layout of Long-Term Care Units: Relationships between Support for Residents’ Well-Being and for Caregivers’ Burnout and Resilience"

_ijerph, 2024, doi:10.3390/ijerph21050575_

Round 1

Reviewer 1 Report

Comments and Suggestions for Authors

The article addresses a very topical subject from an angle that has not been, so far, thoroughly discussed in the literature: the impact of how institutional care facilities are designed on the well-being of residents and staff. The novelty is addressing the impact looking at both groups.

The methodological protocol is sound and clearly outlined. The use of validated scales offers guarantees of validity of findings.

The conclusion that design of spaces impacts the well-being of residents and staff is not novel. But unravelling the apparent paradoxes in the direction of the effects depending on the group is a very critical finding that advances knowledge on what to take into account when designing care facilities. One important idea, that authors do not explicitly address, but that can be read in between lines is that space just by itself will not solve all problems. It may facilitate achieving some results with one group but will be insufficient to solve the problems of another one.

The one topic that is insufficiently addressed is the difference between effects in standard nursing homes and specialised dementia care units. Authors raise some hypotheses to explain the difference but it is a topic that falls beyond the scope of their research. Literature on labour force dynamics in LTC would help shed some light on the subject. Maybe authors should consider adding a note on the need for further insights on this specific topic.

Author Response

Please see below.

Reviewer 2 Report

Comments and Suggestions for Authors

Thank you for the opportunity to review this submission . It is a very well structured study and presented in line with the methods adopted.  This work certainly has relevance internationally. There is one aspect which I think is missing tough, even if mentioned as not been measured and that is use of open spaces, for example in the car home garden. Within many international contexts I have visited including working in a UK context this is often very different.  For example in the UK it is seldom that outdoor spaces are used routinely, where as in Brazil this is common place.  I appreciate weather is a factor so wonder how this relates in your study settings? 

This work links very nicely the wider body on research relating to well being and how you have considered more than one stakeholder well done. 

A more major factor which I couldn't se if the age of your study settings.  For example many care homes in the UK are not purpose built but used from other types of places, such as older Victorian build houses, I wonder how that would translate to these context.  I would be interested in replicating this study in a UK sample for comparison.

Author Response

please see below.

Reviewer 3 Report

Comments and Suggestions for Authors

The article's subject is innovative and pertinent to the exercise of good practice with people in situations of suffering.

Introduction: they refer to nurses and nursing assistants. I think it would be more correct to refer to the differentiated group of professionals integrated into these units (social workers, administrative staff, cleaning, security, etc.). In other words, it gives a complex perspective on the organization of these units.

Research design and methods 95

The authors should give elements of characterization of these units: public, private, installed where, and access to this response so that the international reader can better understand the unit of analysis.

Present a clear conclusion with evidence of results. I emphasize the importance of this article for deepening interdisciplinary knowledge between the different sciences to better provide for the well-being of the person as a person in society.

Author Response

please see below
